A new species of parrot snake, Leptophis (Serpentes: Colubridae) from the Brazilian Cerrado

Albuquerque Nelson R. 1
Martins Roullien H. 1
http://orcid.org/0000-0003-1026-271X Carvalho Priscila S. 1
http://orcid.org/0000-0002-1762-6283 Shepard Donald B. 2
http://orcid.org/0000-0002-8789-3061 Santana Diego J. 1 jose.santana@ufms.br
1 Instituto de Biociências, Universidade Federal de Mato Grosso do Sul , Campo Grande, MS , Brazil
2 Department of Biological Sciences, University of Arkansas at Fayetteville , Fayetteville, AR , United States
Edwards Scott
Electronic publication date: 2025 Jan 30
Publication date: 2025
Volume: 13
Electronic Location ID: e18528
Received 2024 Mar 19; Accepted 2024 Oct 24
Copyright: © 2025 Albuquerque et al.
Copyright year: 2025
Copyright holder: Albuquerque et al.
License: This is an open access article distributed under the terms of the Creative Commons Attribution License, which permits unrestricted use, distribution, reproduction and adaptation in any medium and for any purpose provided that it is properly attributed. For attribution, the original author(s), title, publication source (PeerJ) and either DOI or URL of the article must be cited.
License URL: https://creativecommons.org/licenses/by/4.0/

Keywords: Coloration, Colubrinae, Integrative taxonomy, Neotropical biodiversity, South American dry diagonal

Funding: Coordenação de Aperfeiçoamento de Pessoal de Nível Superior-Brasil (CAPES) 001 Universidade Federal de Mato Grosso do Sul, Ministério da Educação, Brazil Coordination for the Improvement of Higher Education Personnel Capes-PrInt 41/2017–Process 88881.311897/2018–01 Universidade Federal de Mato Grosso do Sul Finance Code 001 Dr. Johnny Armstrong and the School of Biological Sciences at Louisiana Tech University Conselho Nacional de Desenvolvimento Científico e Tecnológico CNPq 402012/2022-4 and CNPq 311284/2023-0 This study was financed by the Coordenação de Aperfeiçoamento de Pessoal de Nível Superior-Brasil (CAPES)-Finance Code 001, and supported by the Universidade Federal de Mato Grosso do Sul, Ministério da Educação, Brazil, and the Institutional Program of Internationalization sponsored by Coordination for the Improvement of Higher Education Personnel (Capes-PrInt 41/2017–Process 88881.311897/2018–01). Roullien H Martins received his fellowship from Coordenação de Aperfeiçoamento de Pessoal de Nível Superior during his graduate studies at Universidade Federal de Mato Grosso do Sul (Finance Code 001). Donald B Shepard received financial support from Dr. Johnny Armstrong and the School of Biological Sciences at Louisiana Tech University. Diego J Santana received research fellowships from Conselho Nacional de Desenvolvimento Científico e Tecnológico (CNPq 402012/2022-4 and CNPq 311284/2023-0). There was no additional external funding received for this study. The funders had no role in study design, data collection and analysis, decision to publish, or preparation of the manuscript.

==============================
We describe a new species of Leptophis (parrot snake) from the Cerrado ecoregion of Brazil. The new species, L. mystacinus sp. nov., differs from all other congeners in the following unique character combination: two Spectrum Green (129) to Light Parrot Green (133) dorsolateral stripes separated by a Buff (5) vertebral stripe, usually continuous onto the tail; loreal scale absent; postocular stripe Jet Black (300), wide and long (up 11 scales long onto nuchal region); maxillary teeth 21–25; ventrals 158–173; subcaudals 141–164; black spots on head absent; supracephalic plates of head not edged with black pigment; adult color pattern lacking dark oblique bands; keels absent on first dorsal scale rows; hemipenis unilobed, noncapitate, with undivided sulcus spermaticus, and first row of hemipenial body with four spines. Phylogenetic analysis of 16S mtDNA sequences indicate the new species is the sister taxon of L. dibernardoi, a species occurring in the neighboring Caatinga ecoregion.

Introduction

The Cerrado of central South America is the world’s most biodiverse savanna (Cardoso Da Silva & Bates, 2002). This ecoregion harbors a rich snake fauna, many of which are endemic (Nogueira et al., 2011). The Cerrado exhibits a remarkable variety of vegetation types, encompassing environments ranging from open grasslands and rocky outcrops to shrublands, and even forested areas like Cerradão and Gallery forests (Eiten, 1972). The ecological and physiognomic variation within the Cerrado has played a significant role in the evolution of its diverse snake fauna (Azevedo, Valdujo & Nogueira, 2016; Bueno et al., 2018). While numerous snake species have been recorded in the Cerrado, studies continue to reveal new species (e.g., Santos et al., 2018; Santos & Reis, 2018; Moraes-da-Silva et al., 2021).

The Neotropical genus Leptophis (parrot snakes) comprises a group of 19 colubrid snakes widely distributed from Mexico through Central and South America (Uetz, Freed & Hošek, 2024). These diurnal serpents inhabit both arboreal and terrestrial environments and have a dorsal coloration ranging from unicolored to striped, or adorned with transverse bands (Albuquerque & Fernandes, 2022). Oliver (1948) conducted the first taxonomic reviews of the genus Leptophis (as Thalerophis), primarily using color pattern to distinguish between species and subspecies, and recognized 12 subspecies of L. ahaetulla. Subsequent studies involving nomenclatural revisions, including the proposal of new scientific names, synonymies, or resurrections (e.g., Peters & Orces‐V, 1960; Henderson, 1976; Harding, 1995; Albuquerque, 2009; Albuquerque, Passos & Gotte, 2012), have contributed to our current understanding and classification of species of Leptophis.

A recent major taxonomic revision based on a thorough examination of meristic, morphometric, color pattern, and hemipenial characters of 1,625 specimens of L. ahaetulla (hereafter referred to as the Leptophis ahaetulla complex) recognized the following 10 taxa: L. ahaetulla, L. bocourti, L. bolivianus, L. dibernardoi, L. coeruleodorsus, L. marginatus, L. nigromarginatus, L. occidentalis, L. praestans and L. urostictus. (Albuquerque & Fernandes, 2022). During a visit to the Butantan Institute in São Paulo, Brazil (23 Feb 2006), the first author noticed a long postocular stripe and a distinct color pattern on the supracephalic and supralabial scales on specimens of live individuals from the Brazilian Cerrado in the state of Tocantins. No further investigation was carried out, which led Albuquerque & Fernandes (2022) to tentatively assign those specimens to L. ahaetulla until a more thorough study of their variation could be undertaken.

Integrative taxonomy, which combines multiple lines of evidence such as morphological, meristic, and molecular data, has become a powerful approach for species description (Padial et al., 2010), especially in complex and diverse groups like snakes (Nascimento et al., 2024; Glaw et al., 2013). In addition, molecular techniques, particularly the analysis of mitochondrial DNA (mtDNA) sequences, allow for the assessment of genetic divergence and phylogenetic relationships among species (Vences et al., 2012). By integrating these methodologies, it is possible cross-validate findings from different data types, increasing the reliability of species description. This comprehensive approach is particularly valuable for morphologically similar taxa, ensuring that new species are accurately identified and described.

As part of an ongoing taxonomic revision of the L. ahaetulla complex (Albuquerque, 2009; Albuquerque, Passos & Gotte, 2012; Albuquerque & Fernandes, 2022), examination of a sample of 14 stripe-patterned individuals of Leptophis revealed the existence of an undescribed species, seemingly endemic to the Cerrado (Vanzolini, 1963). In this study, we employed an integrative approach (Padial et al., 2010) to comprehensively understand the species under investigation. By combining traditional morphological and meristic data with morphometric analyses and molecular techniques, we aimed to provide a throughout perspective on the species identification and differentiation. Such multifaceted approach enabled a cross-validation across different data types, enhancing the robustness and accuracy of our conclusions. Therefore, herein we describe this a new species of Leptophis based on such integrative taxonomic approach.

Materials and Methods

Sampling

The type series of the new species was collected in the Brazilian states of Tocantins and Minas Gerais. Data were collected as previously described in Albuquerque & Fernandes (2022). Specifically, we compared samples of the new species with 1,625 specimens of Leptophis spp., including one of the syntypes (BMNH 1946.1.6.67) of L. bocourti, the holotype (UMMZ 67973) and paratypes (ANSP 11335; CM 23, CM 2702; FMNH 35614-21; MCZ 27553; UMMZ 67974-77, 60701-2, 60709) of L. bolivianus, the holotype (AMNH 9022) and 18 of the paratypes (AMNH 9023-24; ANSP 5182, 18288; CM 6540, 7433; MCZ 11994-95, 12026; USNM 59931-33, 5587, 60598, 15235, 17746, 27821, 27831) of L. coeruleodorsus, the holotype (CHUFC 1104) and paratypes (CHUFC 221, 365, 493, 525, 561, 1140, 1172, 1227, 1244, 1602, 1721, 1722, 1732, 1929, 1980; MCP 17835, 18318; MNRJ 1959–61, 7596; MZUFV 913, 842; MZUSP 23131; MPEG 27110; UFPB 4300; URCA 1245, 5539, 5696, 6127, 6142, 6436, 7890, 9431, 10126, 10617, 10618, 11130, 12023; ZUFMS-REP 3456) of L. dibernardoi, the holotype (AMNH 3531) of L. liocercus, the holotype (ANSP 5514) of L. marginatus, the holotype (BMNH 1946.1.5.7) of L. nigromarginatus, one of the syntypes (BMNH 1946.1.6.62) of L. occidentalis, the holotype (AMNH 17363) of L. ultramarinus, the syntypes (both catalogued as USNM 6754) of L. praestans, the holotype (UMMZ 55528) of L. occidentalis chocoensis (Supplemental Material 1), together with photos of the lectotype of L. ahaetulla (UUZM 2) and literature data (e.g., Oliver, 1948; Albuquerque, Passos & Gotte, 2012; Albuquerque & Fernandes, 2022). Institutional abbreviations are as listed in Sabaj (2023). Vouchers of prepared materials are noted in Supplemental Material 1. Lastly, we counted the number of maxillary teeth of four skulls of L. depressirostris, L. diplotropis, and L. mexicanus to combine with data from Oliver (1948) and Hoyt (1964) to distinguish species of Leptophis.

Variation and sexual dimorphism

We used the following meristic variables in the descriptive analyses of the new species: ventrals, subcaudals, supralabials, infralabials, supralabials entering orbit, infralabials contacting first chin shields, preoculars, postoculars, anterior and posterior temporals, maxillary, palatine, pterygoid, and dentary teeth. We followed the terminology of Dowling (1951) for counting ventral scales. We used the methodology of Harvey & Embert (2008) to describe the variation in these meristic characters. Therefore, sample sizes given for paired characters of pholidosis (e.g., supralabials, infralabials, preoculars) refer to the number of sides examined for all specimens, whereas sample sizes given for other (non-paired) characters (e.g., ventrals, subcaudals) refer to the number of snake specimens examined. We measured the snout-vent length (SVL) and the intact tail length (TL) with a flexible ruler to the nearest 1 mm. We noted specimens with an incomplete tail by adding a “+” to their TL measurement.

We described colors following Köhler (2012), with color names capitalized and color codes in parentheses. We determined the sex of each specimen based on the presence-absence of hemipenes verified through a ventral incision at the base of the tail. We described the hemipenis of the new species based on the right organ prepared from specimen ZUFMS-REP004703, using the method described by Pesantes (1994). Terminology for hemipenial morphology followed Dowling & Savage (1960) and Zaher (1999).

Additionally, we performed a morphometric analysis to investigate interspecific differences using a Random Forest (RF) classification approach. The dataset consisted of measurements from multiple species, including variables such as the number of ventral scales (VE), subcaudal scales (SC), snout-vent length (SVL), and tail length (TL). The RF model was constructed using the R programming language (version 4.4), using the randomForest package (RColorBrewer & Liaw, 2018). We implemented the random forest and generated random classification trees by using bootstrap samples from the original data set to grow 1,000 unpruned classification trees. The model was trained to classify species based on the variables. We calculated variable importance scores to evaluate the contribution of each morphometric trait to the classification. For visualization, the predicted species were plotted against the SVL and TL measurements. To highlight the distribution of each species in morphometric space, we used convex hull polygons to connect the outermost data points for each species. These polygons, along with points representing individual measurements, were visualized using ggplot2 (Wickham, Chang & Wickham, 2016). These analyses were conducted separately for adult males and females of all species, and for the new species and the sister taxa L. dibernardoi and L. marginatus, resulting in a total of four analyses.

Geographical data

We obtained geographical coordinates for relevant specimens in herpetological collections (datum WGS84) or from the online version of the Global Gazetteer (Falling Rain Software, Southampton, PA, USA). We created maps with QGIS. Although the Butantan Institute specimens are not designated as paratypes (see below), we plotted their localities on the map.

Phylogenetic inference and genetic distances

Whole genomic DNA was extracted from muscle or liver tissues of four specimens from the Leptophis ahaetulla complex (one L. ahaetulla, one L. marginatus, and two belonging to the new species) using a Qiagen DNeasy kit (Valencia, CA, USA) following the manufacturer’s protocol. We decided to sequence these four individuals based on the data available in the ZUFMS-REP collection to complement the existing data in GenBank. Next, we amplified a fragment of the mitochondrial 16S gene using primers 16Sar and 16Sbr (Palumbi et al., 1991). The PCR protocol was configured with one initial phase of 94 °C for 3 min, followed by 35 cycles of 94 °C for 20 s, 50 °C for 20 s, and 72 °C for 40 s, with a final extension phase of 72 °C for 5 min. Purification of PCR products and DNA sequencing were performed by Eurofins Genomics Inc. (Louisville, KY, USA).

We combined our newly generated 16S sequences with all comparable 16S sequences of Leptophis deposited in GenBank. In addition, we downloaded from GenBank one 16S sequence of Lampropeltis californiae, Chironius scurrulus and Dendrophidion dendrophis for use as outgroups. Chromatogram sequences were visualized in Geneious v.9.0.5. We aligned the 16S gene fragments using the MAFFT algorithm (Katoh & Toh, 2008) in Geneious v.9.0.5 with default settings. The final alignment comprised 51 sequences of a 404 base-pair (bp) fragment of the mitochondrial 16S gene. All GenBank accession numbers and genetic vouchers used here are listed in Supplemental Material 2.

We performed Bayesian phylogenetic inference in BEAST v.2.7.4 (Bouckaert et al., 2019) using a Yule Process tree prior for 20 million generations, sampling every 2,000 steps. The most appropriate substitution model was GTR+I+G, which was determined using jModelTest (Darriba et al., 2012). We checked for stationarity by visually inspecting trace plots and ensuring all effective sample size values were >200 in Tracer v.1.7.1 (Rambaut et al., 2018). The first 10% of sampled genealogies were discarded as burn-in, and the maximum clade credibility tree with median node ages was calculated with TreeAnnotator v.2.7.4 (Bouckaert et al., 2019). We used this tree in a Generalized Mixed Yule Coalescent (GMYC) analysis to delimit species (Pons et al., 2006; Fujisawa & Barraclough, 2013) with a single-threshold in R v.4.1.1 (R Core Team, 2024) using the package splits (Ezard, Fujisawa & Barraclough, 2017). Finally, we calculated mean pairwise sequence divergences (uncorrected p-distances) among and within species using MEGA v.10.1.1 (Kumar et al., 2018).

Nomenclatural acts

The electronic edition of this article conforms to the requirements of the amended International Code of Zoological Nomenclature, and hence the new names contained herein are available under that Code of this article. This published work and the nomenclatural acts it contains have been registered in ZooBank, the online registration system for the ICZN. The LSID (Life Science Identifier) for this publication is: LSIDurn:lsid:zoobank.org:pub: 9A6A83C3-4068-46AC-9130-B706BB04A0BF. The electronic edition of this work was published in a journal with an ISSN, has been archived, and is available from the following digital repository: www.peerj.com/.

Results

Molecular analysis

Our tree topology (Fig. 1) based on the 16S mtDNA gene recovered the new species as monophyletic (pp = 0.99) and the sister taxon of L. dibernardoi (pp = 0.72). However, most of the clades in the tree are weakly supported, and thus we refrain to further discussion about species relationships. For a better overview of Leptophis phylogeny, we refer to Torres-Carvajal & Terán (2021), in which two mitochondrial and one nuclear gene were used to infer species relationships. Notably, we emphasize the paraphyly observed in L. ahaetulla, consistent with previous studies, suggesting this species comprises a complex of unnamed lineages. The GMYC split the samples analyzed into 27 lineages (confidence interval: [2–30]; likelihood ratio test: 5.22367; result of the LR test: −0.00599) (Fig. 1; Supplemental Material 3). Average sequence divergences between the new species and congeners ranged from 2% (L. dibernardoi) to 8% (L. depressirostris and L. diplotropis) (Table 1).

Figure 1 Gene tree.

Gene tree for the genus Leptophis inferred from Bayesian analysis of the 16S mitochondrial gene fragment. Values adjacent to nodes indicate posterior probabilities. Scale bar represents number of substitutions per site. Grey bars represent each evolutionary entity delimited by the GMYC (Generalized Mixed Yule Coalescent). Photo by L. A. Silva.

Table 1 Average uncorrected (p-distance) sequence divergence for the gene 16S between different Leptophis taxa.

Data in bold on the diagonal are the average intraspecific divergences. n/c = not calculated.

		1	2	3	4	5	6	7	8	9	10	11	12	
1	L. ahaetulla	0.05												
2	L. bocourti	0.04	>0.01											
3	L. coeruleodorsus	0.04	0.04	>0.01										
4	L. cupreus	0.06	0.05	0.05	>0.01									
5	L. depressirostris	0.09	0.07	0.08	0.06	0.01								
6	L. dibernardoi	0.03	0.03	0.03	0.05	0.08	0.01							
7	L. diplotropis	0.08	0.07	0.08	0.09	0.08	0.07	n/c						
8	L. marginatus	0.03	0.02	0.03	0.05	0.08	0.02	0.07	>0.01					
9	L. mystacinus	0.04	0.03	0.04	0.06	0.08	0.02	0.08	0.02	0.01				
10	L. nigromarginatus	0.04	0.04	0.03	0.05	0.08	0.03	0.07	0.02	0.03	0.02			
11	L. occidentalis	0.06	0.05	0.06	0.05	0.08	0.05	0.08	0.04	0.06	0.05	0.05		
12	L. riveti	0.06	0.05	0.06	0.02	0.07	0.06	0.09	0.05	0.06	0.06	0.06	0.01	

Morphometric variation

The RF analysis conducted for males of all species (Fig. 2A) revealed that the most important variables were VE (ventral scales) and SC (subcaudal scales), with L. nigromarginatus and L. marginatus showing particularly high values for SC, at 104.86 and 55.55, respectively. The SVL (snout-vent length) and TL (tail length) variables contributed less to species differentiation, especially in L. sp. nov. and L. coelureodorsus, where lower importance values were observed. Based on the 10-fold cross-validation, the model achieved an overall accuracy of 63.94%. The variable importance scores indicated that VE and SC were key contributors to the classification, while TL was the least influential. In females (Fig. 2B), VE and SC again stood out, with L. nigromarginatus showing the highest VE value (106.18) and L. marginatus having the highest SC importance (42.32). Although SVL and TL contributed less to species differentiation, the model accurately classified species with 62.45% accuracy, and the variable importance analysis confirmed that VE and SC played a significant role in the classification.

Figure 2 Random forest results for species in the Leptophis ahetulla complex.

Variation in ventral scales (VE) and subcaudal scales (SC), the two best predictors of differences among species of the Leptophis ahetulla complex for (A) males, and (B) scale bar of variable importance scores based on mean decrease of guided regularized random forest models. The higher the mean decrease in Gini accuracy, the higher the predictor importance. (C) Confusion matrix showing individual classification error. Variation in VE and SC, the two best predictors of differences among species of the Leptophis ahetulla complex for (D) females, and (E) scale bar of variable importance scores. (F) Confusion matrix showing individual classification error. Variation in VE and SC, the two best predictors of differences among Leptophis mystacinus sp. nov. and its sister species (L. dibernardoi and L. marginatus) for (G) males, and (H) scale bar of variable importance scores. (I) Confusion matrix showing individual classification error. Variation in tail length (TL) and subcaudal scales (SC), the two best predictors of differences among Leptophis mystacinus sp. nov. and its sister species (L. dibernardoi and L. marginatus) for (J) females, and (K) scale bar of variable importance scores. (L) Confusion matrix showing individual classification error. Species abbreviations are the first four letters of the specific epithet shown in (A).

In the analyses restricted to L. mistacynus sp. nov., L. dibernardoi, and L. marginatus, SC consistently emerged as the most important variable for classification. For males (Fig. 2C), SC had values of 19.81 for L. marginatus and 10.74 for L. sp. nov., with other variables, such as VE and TL, having lower contributions. VE showed a value of 3.45 for L. dibernardoi, while TL displayed minimal importance across the species. The model accurately classified species with 72.99% accuracy. For females (Fig. 2D), SC remained the dominant variable, with values of 11.66 for L. sp. nov. and 9.25 for L. marginatus. The contributions of VE and TL were less pronounced, although TL showed a positive value of 5.31 for L. dibernardoi. The model accurately classified species with 79.58% accuracy.

Leptophis mystacinus sp. nov.

(Figs. 3–9)

Figure 3 Holotype of Leptophis mystacinus.

(A) Dorsal and (B) ventral views of the holotype of Leptophis mystacinus (ZUFMS-REP004702), from Pium, state of Tocantins, Brazil. Photo by D. J. Santana.

Figure 4 Holotype of Leptophis mystacinus (ZUFMS-REP004702).

(A) Right and left (B) lateral views of head of the holotype of Leptophis mystacinus (ZUFMS-REP004702) in life, from Pium, state of Tocantins, Brazil. Photos by L. A. Silva.

Figure 5 Holotype of Leptophis mystacinus.

(A) Dorsal, (B) ventral, and (C) lateral views of the head of the holotype of Leptophis mystacinus (ZUFMS-REP004702), from Pium, state of Tocantins, Brazil. Photo by D. J. Santana.

Figure 6 Comparative head morphology in the holotypes of the Leptophis species from the South American dry diagonal.

(A) Dorsal and (B) lateral views of Leptophis mystacinus sp. nov., (ZUFMS-REP 4702); (C) dorsal and (D) lateral views of L. marginatus (AMNH 5514), and (E) dorsal and (F) lateral views of L. dibernardoi (CHUFC 1104).

Figure 7 Comparative coloration in life among the Leptophis species from the South American dry diagonal.

(A) Leptophis mystacinus sp. nov., (ZUFMS-REP 4702) from Pium, Tocantins, Brazil, (B) L. marginatus from Corumbá, Mato Grosso do Sul, Brazil, and (C) L. dibernardoi from Macaíba, Rio Grande do Norte, Brazil. Photo credit: L. A. Silva (A), S. Keuroghlian-Eaton (B) and W. Pessoa (C).

Figure 8 Paratype of Leptophis mystacinus sp. nov.

Left lateral view of head of the paratype of Leptophis mystacinus (MNRJ 6672) from Várzea da Palma, Minas Gerais. Photo by R. Rodrigues.

Figure 9 Female Leptophis mystacinus.

Female Leptophis mystacinus sp. nov. (MNRJ 6672, SVL 595 mm, TL 375+ mm) from Várzea da Palma, Minas Gerais, exhibiting the banded pattern typical of juveniles of most species of Leptophis. Photo by R. Rodrigues.

Leptophis ahaetulla—Albuquerque, Passos & Gotte, 2012: 249 (in part); Murphy et al., 2013: 563 (in part); Torres-Carvajal & Terán, 2021: 3 (in part); Albuquerque & Fernandes, 2022: 11 (in part).

Leptophis liocercus—Albuquerque, Passos & Gotte, 2012: 248 (in part); Albuquerque & Fernandes, 2022: 28 (in part).

Holotype. ZUFMS-REP004702, a female collected 29 November 2017 by L.A. Silva, R.M. Fadel and H. Folly at municipality of Pium, Instituto Araguaia (10°26′34″S, 49°10′55″W; datum = WGS 84), state of Tocantins, Brazil.

Paratypes (n = 13). Minas Gerais, Várzea da Palma, Fazenda Corrente: MNRJ 56672, female, collected 16–18 November 1987 by G. Kiteumacher & M. Porto. Tocantins, Araguaína, Parque Urbano Ecológico Cimba, ZUFMS-REP 4700, male, collected 30 March 2017 by S.P. Dantas; Arraias, UHC Pau D’arco, ZUFG 885, female, collected October 2008 by R.M. Oliveira; Palmas: CHUNB 23620, female, date of collection and collector unknown; Peixe: CHUNB 52568, female, collected 20 April 2006 by G.R. Colli, CHUNB 52569, female, collected 20 April 2006 by G.R. Colli, CHUNB 52570, male, collected 20 April 2006 by G.R. Colli, CHUNB 52571, male, collected 20 April 2006 by G.R. Colli, CHUNB 52572, male, collected 20 April 2006 by G.R. Colli, CHUNB 52573, female, collected 20 April 2006 by G.R. Colli, CHUNB 52598, female, collected 20 April 2006 by G.R. Colli; Pium: CHUNB 24750, male, collected 11 September 2001 by F.G.R. Franca and G.H.C. Vieira; ZUFMS-REP 4703 (Instituto Araguaia), male, collected 28 January 2018 by L.A. Silva, R.M. Fadel and H. Folly.

Referred specimens. IBSP 64270 (Palmas, U. H. Luís Eduardo Magalhães), 64396, 64514, and 65907 (Lajeado, U. H. Luís Eduardo Magalhães). All of these specimens were deposited in the Herpetological Collection Alphonse Richard Hoge of Instituto Butantan (IBSP), São Paulo, Brazil (partially and tragically destroyed by fire on 15 May 2010).

Diagnosis

Leptophis mystacinus sp. nov. can be distinguished from all currently recognized congeners by a unique combination of the following characters: two Spectrum Green (129) to Light Parrot Green (133) (Sky Blue (167) in preservative) dorsolateral stripes (2–4 scales wide, at least anteriorly) separated by a Buff (5) (Light Sky Blue (191) in preservative) vertebral stripe (1–1.5 scales wide), usually continuous onto the tail (occasionally indistinct on posterior third of tail); dorsal scale rows below the lateral stripes usually Dark Spectrum Yellow (78) (Sky Blue (167) in preservative); loreal scale absent; postocular stripe Jet Black (300), wide (extending to lower postocular, lower half to two-thirds of anterior temporal, one-third to lower half of lower posterior temporal, upper edges of last three supralabials) and long (up 11 scales long onto nuchal region); anterior to orbit, stripe reduced to black margin of supralabials 1–3 or 1–4, posterior lower edge and anterior upper edge of nasal and upper edge of rostral scale. Ventral surfaces of head, trunk, and tail white to Smoky White (261). Maxillary teeth 21–25; ventrals 158–166 in males, 158–173 in females; subcaudals 153–164 in males, 141–158 in females.

Comparisons

Leptophis mystacinus sp. nov. is similar to L. ahaetulla and L. dibernardoi in its dorsal coloration, with all specimens examined sharing the pattern of two green dorsolateral stripes separated by a vertebral stripe, at least anteriorly (Fig. 3). However, in life, the second (on the anterior region of trunk) to fourth (middle to posterior region) dorsal scale rows are Dark Spectrum Yellow (78) in the new species (Fig. 4) (vs. second to fourth rows Sulphur Yellow (80) in L. ahaetulla and White to Pale Sulphur Yellow (92) in L. dibernardoi). The Jet Black (300) postocular stripe is narrower in L. mystacinus sp. nov. than in L. dibernardoi, occupying lower one-third to half of lower posterior temporal (vs. most of the lower posterior temporal pigmented in L. dibernardoi), and wider than in L. ahaetulla (the latter with lower edge to one-third of lower posterior temporal pigmented). The postocular stripe Jet Black (300) is longer in L. mystacinus sp. nov. (Figs. 5–7) than in L. ahaetulla and L. dibernardoi, extending up to 11 scales posterior to last supralabial (vs. postocular stripes not extending beyond two scales onto nuchal region in L. ahaetulla and up to four scales posterior to last supralabial in L. dibernardoi); anterior to orbit, stripe reduced to black margin of supralabials 1–3 or 1–4, occupying posterior lower edge and anterior upper edge of nasal and upper edge of rostral scales in L. mystacinus sp. nov. (vs. nasal and rostral scales immaculate in L. ahaetulla and usually immaculate in L. dibernardoi). The first row of the hemipenis bears four spines in L. mystacinus sp. nov. (vs. 5–8 spines in L. ahaetulla and 8–9 in L. dibernardoi). Another species found in Cerrado, L. marginatus, does not exhibit the pattern of two dorsolateral stripes separated by a vertebral stripe (Figs. 6, 7).

Description of holotype (Figs. 3–5)

Adult female, SVL 475 mm, TL 323 mm (68% of SVL); body cylindrical with flattened belly, angulate paraventral region. Head elongate, distinct from neck, wider than midbody diameter; head length 16.10 mm (3.4% of SVL), head height 4.5 mm, head width 8.3 mm; maxillary teeth 25/25; rostro-orbital distance 5.25 mm; rostral wider than high, visible in dorsal view; nasals undivided, separated from preocular by prefrontal; prefrontal in contact with supralabials 2–3; prefrontals contacting supraocular, slightly larger than internasals; prefrontal width/snout length ratio 0.87; frontal triangular, longer than wide, more than twice as long as the prefrontal; parietals length (5.93 mm) greater than width (3.89 mm), contacting upper and lower postoculars; two postoculars, lower one slightly smaller than the upper one; right temporals 1 + 2, anterior one contacting parietal, lower postocular, supralabials 7–9, upper posterior temporal contacting parietal; left temporals 1 + 2, anterior one contacting parietal, lower postocular, supralabials 7–8; upper posterior temporal contacting parietal; orbit 3.27 mm long, smaller than snout length; eye large, pupil round; preocular single, contacting supraocular, frontal and prefrontal; supralabials 8/8, 4–5/4–5 contacting orbit; infralabials 10/10, first five contacting anterior chinshields; first pair of infralabials in contact behind symphysial, preventing symphysial/chinshield contact; chinshields in two pairs, elongate, separated by mental groove; posterior chinshields longer than anterior ones; ventrals 169; scales on first row slightly larger than those of adjacent series; dorsal scale rows 15/15/11; weak keels on dorsal scales of trunk, except the first scale row of each side, which are smooth; cloacal shield divided; subcaudals in 165 pairs; keels on ventrals and subcaudals, weaker on subcaudals; dorsal scales of tail smooth; single apical pit present on all dorsal scales of trunk except first dorsal row. Dorsum of head Venetian Blue (170); narrow Jet Black (300) postocular stripe occupying lower postocular, anterior portion and lower half of anterior temporal, two-thirds of lower posterior temporal, and upper edges of last three supralabials extending to the 9th scale row posterior to ultimate labial on each side; anterior to orbit, stripe reduced to black margin of supralabials 1–5, posterior lower edge and anterior upper edge of nasal, and upper edge of rostral scales; supralabials Pale Buff (1), except for upper margin of those under ocular stripe; anterior lower margin of nasal and lower one-half to two-thirds of rostral Pale Buff (1); vertebral stripe Light Sky Blue (191) covering scales from 25th vertebral scale to end of trunk; two dorsolateral Sky Blue (167) stripes, separated from each other by vertebral stripe, extending from 25th vertebral scale to tip of tail; blue stripes becoming indistinct on anterior third of tail; ventral surfaces of head, trunk, and tail Pale Buff (1).

Variation

Largest male (ZUFMS-REP004703) SVL 738 mm, TL 500 mm; largest female (CHUNB 23620) SVL 868 mm, TL 522+ mm; ventrals 158–166 in males ( x¯ = 160.8 ± 3.6, n = 5), 158–173 in females ( x¯ = 167.9 ± 4.7, n = 9); subcaudals 153–164 in males ( x¯ = 159.8 ± 4.8, n = 4), 141–165 in females ( x¯ = 154 ± 9.1, n = 6); the tip of tail is mutilated in CHUNB 23620, CHUNB 24750, MNRJ 56672 and ZUFG 885. This is why we did not record the number of subcaudal scales and tail length for these specimens; supralabials 8 (n = 19), 9 (n = 8), or 7 (n = 1), with fourth–fifth (n = 20) or fifth–sixth (n = 8) entering the orbit; infralabials 10 (n = 19), 9 (n = 7), or 11 (n = 2), with first five (n = 21), four (n = 5), or six (n = 2) contacting first pair of chinshields; preoculars 1 (n = 28); postoculars 2 (n = 27) or 3 (n = 1); anterior temporal 1 (n = 28); posterior temporal 2 (n = 24) or 1 (n = 4).

In CHUNB 24750, postocular stripes extend to the 10th scale row posterior to ultimate supralabial on each side. Specimen ZUFMS-REP004700 has a distinct postocular stripe covering the lower postocular, lower half of anterior temporal, lower margin of lower posterior temporal, and upper edges of last three supralabials extending to the 11th scale row posterior to ultimate labial on each side. Specimen ZUFMS-REP004703 has a postocular stripe reduced to black margin on lower postocular, occupying lower half of anterior temporal, two-thirds of lower posterior temporal, and upper edges of last three supralabials extending to the 7th scale row posterior to ultimate labial on each side. Specimen ZUFG 885 has a postocular stripe covering the lower margin of upper postocular, upper half of lower postocular, lower half of anterior temporal, lower margin of lower posterior temporal, and upper edges of last three supralabials extending to the 7th scale row posterior to ultimate labial on left side. Specimen MNRJ 6672 (Fig. 8) has a postocular stripe covering the lower half of upper postocular, upper half of lower postocular, two-thirds of anterior temporal, lower margin of upper posterior temporal, nearly all of posterior temporal, and upper edges of last three supralabials extending to the 9th scale row posterior to ultimate labial on left side. Specimens MNRJ 6672 (Fig. 9), a female of 970 mm in total length (but tail incomplete), and ZUFG 885, a female of 605 mm in total length (but tail incomplete), are ornamented with bands in anterior and middle region of body, similar to those found in juveniles of other species of Leptophis (see Oliver, 1948; Albuquerque & Fernandes, 2022).

Hemipenial morphology (Fig. 10)

Retracted organ extends to level of subcaudal 5; fully everted and almost completely expanded hemipenis renders a unilobed and noncapitate organ; sulcus spermaticus undivided with centrolineal orientation, extending from base to distal tip of organ; basal region of hemipenial body with numerous spines distributed in five rows; spines arranged irregularly rather than in transverse rows; first row with four spines; two spines on the first row adjacent to sulcus spermaticus larger than those of other rows; three spinules on asulcate side of basal region, occurring between first row of spines; small spines gradually becoming stout papillae on median region of hemipenial body; calyces poorly developed on distal portion of hemipenial body, with barely developed papillae; most central portion of lobe nude; asulcate side similar to sulcate side.

Figure 10 Hemipenis of Leptophis mystacinus.

(A) Sulcate and (C) asulcate sides of the hemipenis of Leptophis mystacinus (ZUFMS-REP004703), from Pium, state of Tocantins, Brazil. Photo by D. J. Santana.

Distribution and natural history

Leptophis mystacinus is known from the Brazilian states of Tocantins and Minas Gerais. Based on the localities associated with voucher specimens, L. mystacinus occurs in areas inside the Cerrado (Fig. 11) as well as regions influenced by the humid Amazon rainforest (e.g., Araguaína, Caseara and Pium, all in Tocantins state). The specimen CHUNB 52572 (SVL 979 mm) contains four well-developed eggs (the first, along head-tail orientation, measured 30.95 mm).

Figure 11 Geographic distribution of Leptophis mystacinus and related species.

Geographic distribution of Leptophis mystacinus sp. nov., L. marginatus and L. dibernardoi mapped onto South American ecoregions. DJ Santana prepared the map using QGIS 3.8. Base maps were obtained from IBGE (https://www.ibge.gov.br/geociencias/downloads-geociencias.html) and Ecoregions 2017 (https://ecoregions.appspot.com/).

Etymology

The specific name is derived from the Greek mystax (transliteration of μύσταξ), meaning ‘upper lip’ or ‘mustache’, and the Latin suffix -inus, denoting ‘likeness’ or ‘belonging to’. The black pigmentation covering the rostral scale of Leptophis mystacinus is distinct in most individuals, giving the appearance of a mustache.

Discussion

Albuquerque & Fernandes (2022) highlighted the difficulty of differentiating species of the Leptophis ahaetulla complex based only on meristic characters. This is particularly the case for the number of ventral and subcaudal scales, as these characters exhibit considerable interspecific overlap, underscoring the importance of characters related to color pattern for recognizing species in the L. ahaetulla complex. Some specimens from Tocantins examined by these authors and herein described as L. mystacinus, for example, exhibit a similar dorsal color pattern to L. ahaetulla and L. dibernardoi (all three with dorsolateral stripes), although the combination of the general color pattern and the rostral scale distinctly edged with black unequivocally distinguishes L. mystacinus from these other two species. Long postocular stripes can also be found in other species of Leptophis such as L. bolivianus, L. coeruleodorsus, and L. liocercus, and are even considered an unstable character within the genus (Oliver, 1948; Albuquerque, Passos & Gotte, 2012), but they are important for distinguishing among species occurring in the South American diagonal of open formations (i.e., L. ahaetulla, L. dibernardoi, L. marginatus, L. mystacinus).

The South American diagonal of open formations, as proposed by Vanzolini (1963), delineates a significant ecological corridor encompassing the Caatinga, Cerrado, and Chaco. This diagonal exhibits notable biotic similarities across its ecoregions, harboring many widespread species and many sister-taxa pairs in neighboring ecoregions (Rodrigues & Prudente, 2011; Werneck, 2011; Ledo et al., 2020; Oliveira et al., 2015). In our present study, we unveiled a new species of Leptophis within the Cerrado region, which we inferred to be the sister taxon of L. dibernardoi, a species typically found in the Caatinga and bordering Atlantic Forest regions (Albuquerque et al., 2022). We used the 16S mtDNA, usually used for barcoding reptiles (Vences et al., 2012), to diagnose the proposed new species by analyzing genetic divergence and constructing phylogenetic relationships to ensure its monophyly. This molecular approach provides robust and objective criteria to support the recognition of the proposed new species alongside traditional morphological methods (Miller, 2007). On the other hand, the GMYC identified 27 evolutionary entities, splitting several single species, which could indicate either cryptic diversity within the genus or an oversplitting in Leptophis. Based on genetic distance, the new species is closest to L. dibernardoi and L. marginatus (2%), both species found within the South American dry diagonal ecoregion. Although L. marginatus has not been recovered as closely related, it is worth noting that the nodes are weakly supported. A more comprehensive multi-locus dataset could potentially provide stronger support for relationships in the genus as showed in Torres-Carvajal & Terán (2021), which presented a molecular phylogeny and addressed the systematics in the genus.

Despite successfully delimiting the new species Leptophis mystacinus as monophyletic, the low support for several deeper nodes in the tree highlights the limitations of using a single mitochondrial DNA locus (16S) for phylogenetic reconstruction. While our findings provide a framework for understanding species boundaries, a more robust phylogeny of the genus would benefit from multilocus approaches or high-throughput sequencing (HTS) techniques, which could better resolve the evolutionary relationships and offer stronger support for deeper divergences. The non-monophyly of Leptophis ahetulla, L. nigromarginatus or even L. occidentalis in the phylogenetic tree suggests that these taxa may represent complexes of cryptic species rather than a single evolutionary lineage. This indicates a potential for the discovery of additional new species within Leptophis. Therefore, further taxonomic revisions, combined with comprehensive morphological and ecological data, are essential to delineate the species boundaries and assess the full extent of diversity within the group.

The congruence of species distributions and ecoregions is not unexpected, given the distribution patterns observed in other squamate species (Santos et al., 2012). For example, the Cerrado endemic lizard Vanzosaura savanicola is the sister species of V. multicustata found in the Caatinga (Recoder et al., 2014). A comprehensive biogeographic study conducted on a broader scale would be fundamental in elucidating the ancestral origins and diversification of Leptophis. Such an endeavor holds the potential to unravel the evolutionary history of Leptophis and contribute significantly to our understanding of Neotropical biodiversity.

Supplemental Information

Supplemental Information 1 Fasta alignment.

16S mtDNA alignment for Leptophis and outgroup taxa used in the present study

Supplemental Information 2 Specimens examined.

Asterisk denotes specimens for which we also examined the hemipenes (*) or skulls (**). In some cases we only examined the skulls (***). We write the country and state names in capital letters and separate the locality names with commas and the state names with a semicolon.

Supplemental Information 3 GenBank accession numbers for sequence data, specimen voucher numbers, collecting locality information, and references for all samples included in molecular analyses.

Supplemental Information 4 GMYC results.

Tree with the GMYC results. Branches connected in red belong to the same evolutionary entity.

We are grateful to all curators, collections manager and their respective institutions for the loan of specimens and/or for permission to examining specimens in their care. NA thanks his advisors, M. Di Bernardo (I. memoriam) and T. de Lema (I. memoriam) for their support and encouragement throughout his graduate studies. NA is indebted to D. R. Frost, D. Kizirian, T. Grant as well as the whole team of the Department of Herpetology at the AMNH for the opportunity to develop part of his thesis under their supervision and support. We thank H. C. Costa for helpful insights on the etymology and with the vernacular names.

Additional Information and Declarations

Competing Interests

The authors declare that they have no competing interests.

Author Contributions

Nelson R. Albuquerque conceived and designed the experiments, performed the experiments, analyzed the data, prepared figures and/or tables, authored or reviewed drafts of the article, and approved the final draft.

Roullien H. Martins performed the experiments, analyzed the data, prepared figures and/or tables, authored or reviewed drafts of the article, and approved the final draft.

Priscila S. Carvalho performed the experiments, analyzed the data, authored or reviewed drafts of the article, and approved the final draft.

Donald B. Shepard performed the experiments, analyzed the data, authored or reviewed drafts of the article, and approved the final draft.

Diego J. Santana conceived and designed the experiments, performed the experiments, analyzed the data, prepared figures and/or tables, authored or reviewed drafts of the article, and approved the final draft.

DNA Deposition

The following information was supplied regarding the deposition of DNA sequences:

All the sequences are available at GenBank:

HM582222, MK209302, KF667667, KF667670, KF667671, MH140827, MH140828, MH140829, MZ126814, MZ126815, MZ126816, MZ126817, MZ126818, MZ126819, MZ126820, MZ126821, MZ126822, PP463857, KF667666, KF667672, KF667673, KF667674, MZ126823, MZ126824, KR814643, KX660270, MH140830, MH140831, MH140832, MZ126825, MZ126826, MZ126827, MZ126828, ON123616, ON123618, KX660271, KF667675, MN276246, MN276247, MN276248, KF667669, KF667668, MZ126829, MZ126830, MZ126831, KU323980, KX660266, MK086632, PP463855, PP463854, PP463856, PP463857.

Data Availability

The following information was supplied regarding data availability:

The raw data, 16S mtDNA alignment and R script are available at GitHub:

- https://github.com/Rhinella85/Leptophis_mystacinus_sp_nov.

- Diego J. Santana. (2024). Rhinella85/Leptophis_mystacinus_sp_nov: Albuquerque et al. 2024-Leptophis mystacinus sp. nov. (v1.0). Zenodo. https://doi.org/10.5281/zenodo.13940783.

New Species Registration

The following information was supplied regarding the registration of a newly described species:

Leptophis mystacinus LSID: urn:lsid:zoobank.org:act:FA2FB4B6-5F13-4D17-AF59-B797FAF8F709

Publication LSID: urn:lsid:zoobank.org:pub:9A6A83C3-4068-46AC-9130-B706BB04A0BF.

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
