# Peer review of "A new species of parrot snake, Leptophis (Serpentes: Colubridae) from the Brazilian Cerrado"

_PeerJ, doi:10.7717/peerj.18528_

## Round 0.1 · original submission · Major Revisions

The two reviewers both find the manuscript technically accurate and agree that the new taxon is likely valid. But both reviewers feel that the authors need to be more explicit about the criteria they are using to delimit species. Furthermore, both reviewers suggest additional details and clarity on the samples and the process of delimitation. We look forward to your revisions.

Reviewer 1 ·

Basic reporting

There are several places I suggest slight wording/phrasing revisions, but there are not any problems overall with the writing, how the species is described, and the figures are well done. The datasets are included for molecular data.

Experimental design

The overall design fits within the boundaries of typical species descriptions. I have general comments on how this can be improved overall for the structure of the ms and to better justify the description of the new taxon.

Validity of the findings

The authors provide all relevant data and conclusions are justified given this data. I provide some comments on additional points the authors may want to consider in discussing the new taxon.

Additional comments

Dear Authors,

I enjoyed reading this ms, and always am interested in learning about a new snake species, especially one from such a charismatic genus, the parrot snakes. The description of the new species, formally, is thorough and well done. And I think the figures all support the text and look good. However, I have some suggestions to improve the overall ms prior to its publication.

The inclusion of general results could be greatly improved, as I detail below. The discussion is also very short and I think it would be worth considering what else might be included there to make it more of a discussion. In particular, I think the authors should consider discussing some of the pros and cons of mtDNA single locus descriptions; to be clear, I do not think this is a negative thing, but I do think it is worth a discussion on the strengths and weaknesses of the approach and justification of it here, it is almost touched on at line 308 but not enough. There is a lot to think about when it comes to the difficulty of species descriptions for the genus in question (which is mentioned in the start of the discussion and very nicely included), snakes in general, and for the Neotropics as a whole.

The DNA dataset and the species description/naming follows the appropriate guidelines. However, PeerJ recommends that you include the new species name in the title of the paper, as well as the abstract. The authors should consider adding the name to the title and/or abstract to follow this suggestion (“In order to increase discoverability, when naming a new species we recommend that the names are mentioned in the title (where feasible) and in the abstract of your article.”). Something to consider, but not strictly necessary in my opinion. I would defer to the AE for this.

In addition, I do not know if this gets added in later during the proofing stage, as I saw the statement elsewhere from the authors for this information, but in the actual text of the ms, this was missing
“- Authors must use the following text in the Methods section: "The electronic version of this article in Portable Document Format (PDF) will represent a published work according to the International Commission on Zoological Nomenclature (ICZN), and hence the new names contained in the electronic version are effectively published under that Code from the electronic edition alone. This published work and the nomenclatural acts it contains have been registered in ZooBank, the online registration system for the ICZN. The ZooBank LSIDs (Life Science Identifiers) can be resolved and the associated information viewed through any standard web browser by appending the LSID to the prefix http://zoobank.org/. The LSID for this publication is: [INSERT HERE]. The online version of this work is archived and available from the following digital repositories: PeerJ, PubMed Central SCIE and CLOCKSS."

It is not clear to me when I start reading the methods where the new species samples come from, as it starts off by mentioning the prior study of Leptophis and the comparisons. But I would expect a section on sampling to start by saying something about the samples of the new species specifically being noted as potentially a new spp due to the striping mentioned in the intro etc., where they all came from, and also refer to the table of samples as well at that point so it’s obvious what those samples are and how they ended up being chosen here.

At line 113, the molecular work is introduced but why/how were the four individuals chosen here?

What is the criteria, a priori, being used by the authors to delimit species in general here? It’s worth being explicit for the readers so that there is no question on this. Integrative methods are mentioned in the intro but are kept vague and there’s no further discussion of just how the species is determined to be valid under a set of specific criteria. For example, some discussion of what are typical mtDNA divergence %s between other species in the genus and how the new species compares could be better included. Table 1 has this info and I thought it was pretty interesting just how low some of the pairwise distances were between some of these species! Not the new species in particular, but just in general. Makes me want some additional info for some of the previously described taxa…! (to be clear not for the scope of this paper, just because of my interest in these species).

I find the results jumping directly to the species description vs some of general results of the work/phylogenetics to be abrupt. I would expect the results to start off reporting as to what was found that would warrant a species diagnosis/description and then go to the description. The results, given that a phylogeny was inferred and that specimens were examined, isn’t just a species description, there is more to it than that to be reported prior to the description of the species itself. I would greatly revise the order of the results. I spent some time looking at other species descriptions in PeerJ, so here are some examples that are more along the lines of what I would expect (note I am not suggesting your formatting follows any of these exactly but they all illustrate providing general results prior to the species description and discussion): https://peerj.com/articles/2767/ ; https://peerj.com/articles/15399/ ; https://peerj.com/articles/17175/ ; https://peerj.com/articles/15195/ ;

Some minor comments on where the phrasing might be reconsidered/revised:

Line 29 “populations concentrated there”
Line 34, I’d make this two sentence paragraph part of the previous paragraph
Line 36/37, this sentence doesn’t really add anything, either cut it out or describe what you mean/what kinds of evidence you area talking about, this is too vague
Line 39 “colubrine” but paper also have the family in parentheses next to it
Line 51 and throughout----do you need to keep spelling out the genus? Sometimes you do and sometimes you do not, and typically after introducing it in each section, you would not write it in full again unless it starts a sentence or could be ambiguous with another taxon.
Line 62, I would say “seemingly” instead of “apparently”
Line 63, heminial? Should this be hemipenal?
Line 104, “hemipenis” should probably be the plural here, even if it is only the right one being described it represents both
Line 109, revise this as there are more commas than needed

Reviewer 2 ·

Basic reporting

The manuscript describes a new species of Leptophis from the Brazilian Cerrado. Overall, the text is clear, unambiguous, and technically correct. The manuscript structure, figures, and tables are of professional quality. Literature references and field background, however, are lacking. For instance, authors should indicate what they mean by "DNA barcoding" and how they used this approach to aid in delimiting and diagnosing the proposed new species.

The Introduction and Materials and Methods sections should detail how authors adopted an integrative approach. For instance, how can phylogenetic inference and genetic distances inform species delimitation? Moreover, the raw data available in the manuscript is rather incomplete. Authors should provide the full alignment and the meristic and morphological data of all 1625 examined specimens.

The figures need many improvements to illustrate differences between sympatric or closely related species and the geographic distribution of species that occur in the region. Authors should refrain from directing readers to other articles to obtain this information.

Experimental design

The authors mentioned that they used data from 1625 examined specimens. However, they present no quantitative analysis using the data, and the data is not available. Authors indicate the difficulty of differentiating species in this genus using meristic data but provide no compelling evidence for such a claim. Multivariate analyses are badly needed to either support the claim that meristic data are useless in species diagnoses in the group or to reveal the relevant characters for species delimitation.

The molecular analysis was based exclusively on 16S data, while data on several other markers are available from a previous publication (Torres-Carvajal, O. & Terán, C. (2021). Molecular phylogeny of Neotropical Parrot Snakes (Serpentes: Colubrinae: Leptophis) supports underestimated species richness. Molecular Phylogenetics and Evolution 164, 7). The phylogenetic analysis should include all available data to produce a robust and up-to-date inference.

Validity of the findings

Despite the availability of molecular data and an impressive sample of meristic and morphological data, no actual quantitative analysis is used to support the delimitation and recognition of the proposed new species, except for the fact that the samples form a monophyletic group and that the average sequence divergences between the proposed new species and congeners range from 2% to 9%. Considering all the available data and the scarcity of analyses and results, the evidence for a new species is unconvincing.

Additional comments

Additional comments are in the attached pdf file.

Annotated reviews are not available for download in order to protect the identity of reviewers who chose to remain anonymous.

Reviewer 3 ·

Basic reporting

.

Experimental design

.

Validity of the findings

.

Additional comments

Major concerns:

1) It certainly is not any reviewer’s role to impose the choice of methods on the authors, but I strongly suggest using statistical approaches when analyzing morphological data.
In this study, there is no proper analysis of the morphological evidence, and the presentation of the comparative analysis is poor, preventing a thorough evaluation of the trait used to diagnose the addressed species/OTUs. There is no table or graph illustrating the distribution of the morphological variation used in this study.
The manuscript will improve considerably if the authors provide statistical analyses, graphs and tables to help the readers better assess the differences among taxa within the L. ahaetulla species complex.

2) The authors do not make the morphological dataset freely available. Please, deposit your morphological dataset in any of the available online depositories (e.g., figshare, dryad, etc). Currently, this is a standard scientific procedure… even for taxonomists.

3) Please, check the numbers regarding the analyzed material (specimens). They do not match throughout the text, particularly with the numbers mentioned in the species description (skulls, hemipenes, etc).

4) Although including several sequences generated by Torres-Carvajal & Terán (2021), the authors did not comment on the striking differences between their resulting tree topology and the one provided in that study.
It is essential that the authors address and compare their findings with contradictory results from other studies. Failing to do that may raise concerns about whether they deliberately avoided discussing contradictory evidence.
The 16S matrix of Carvajal & Terán (2021) is very similar to the molecular matrix analyzed in this MS, although they have included additional data for two other genes (cytb and PRLR). Regarding gene sampling, the study of Carvajal & Terán (2021) is more comprehensive than the MS presented here. I cannot understand why the authors did not sequence at least cytb for the two individuals of the new species. By sequencing at least one more gene, the authors could provide a better test of their phylogenetic hypothesis considering the contradictory evidence.
The comparison between these two studies is crucial for the discussion of the taxonomy of the L. ahaetulla species complex. In Torres-Carvajal & Terán (2021), the species L. nigromarinatus is non-monophyletic, including the genetic diversity of L. coeruleodorsus, L. dibernadoi and L. mystacinus sp. nov.
Moreover, the authors reidentified several samples sequenced in other studies, without providing any formal explanation or indication that they had access to those individuals.

5) The Introduction and Discussion sections are poorly developed. The widespread paraphyly of L. ahaetulla must be properly discussed, indicating which populations can be morphologically differentiated from others, and which populations are morphologically indistinguishable. This detailed discussion is crucial for understanding the taxonomic and evolutionary implications of the findings.


General Comments

1) The authors described the evaluation of a large sampling (1625 specimens) to gather morphological data for comparative analysis. As far as I understand, this morphological dataset was built somewhere else (Albuquerque 2009; Albuquerque et al. 2012; Albuquerque & Fernandes 2022). Which part of this dataset was generated specifically for this study? The authors need to be clear about the study contribution regarding data collection.

2) A map illustrating the distribution of all other species of the complex would be much appreciated. This will allow the readers to evaluate the limits and overlaps among the distribution of the new species and other related taxa.

3) In line 45, the authors have cited only one species as an example of synonymy. It would be more consistent to either include all relevant species names or refrain from citing any specific examples of synonymy. This approach would ensure clarity and consistency in the presentation of taxonomic information.

4) The authors mentioned the evaluation of 58 hemipenis and 30 skulls of several species of Leptophis. However, the MS does not provide any comment about the comparison of this material. There is no illustration of such impressive morphological material (only a hemipenis of a single specimen was illustrated). The authors must conduct a proper evaluation of their morphological dataset. I cannot see the reason for preparing all this fine material without making it useful for your study and for the herpetological community.

5) The authors mentioned in lines 68-82 that they analyzed type specimens of various species related to the new species. This alone represents a significant contribution to the study and to the herpetological community. However, no results or discussion regarding this material were provided. I suggest a proper evaluation of the morphology of the type specimens of all synonyms of Leptophis ahaetulla—especially those species occurring in the Cerrado—using statistical approaches to test and compare these differences. Alternatively, you should at least graphically show the distribution of the morphological variability (using uni- or multivariate graphs) and the relative position of the type species.

6) Please, provide diagnostic characteristics for both males and females of the new species. This will ensure that the study adheres to the general standards of the herpetological community.

7) Table 1 describes the genetic distance of L. cf. ahaetulla related to other species. However, the definition of L. cf. ahaetulla was not addressed in the text. The authors should provide a clear definition of what they consider L. cf. ahaetulla. This will prevent future taxonomic confusion, ensuring consistency in the interpretation of their genetic data.

8) The authors mention the merits of “integrative taxonomy”, even including this term in the MS keywords. However, this MS does not provide any kind of integration between different sets of evidence. The genetic data was probabilistically analyzed, while the morphological data was merely described. The authors should align their methods of analysis for both phenotypical and genetical evidence if they wish to adhere to the integrative taxonomy approach.

4) I have found several typos (e.g., line 63 “heminial”) and unclear/odd sentences (e.g., line 307-308 “…it's worth noting that the nodes...”). I did not indicate them in the text because it is my opinion that your MS will considerably change if you agree with my comments and suggestions. Please, provide a careful proofreading of your text before submitting a revised version of your MS.

Final remarks

I would like to clearly state that I consider this study an important contribution to South American herpetology and another significant piece in the puzzle for understanding the biodiversity patterns of the SA open diagonal. Therefore, I encourage the authors to carefully consider my comments and suggestions. The morphological dataset you have is impressive and deservers a proper treatment. I am confident that, beyond just describing another species with few comments about the complex, you have the opportunity to make a greater contribution to our field.

---

## Round 0.2 · Minor Revisions

I have decided to give a recommendation of minor revision even though reviewer 2 has extensive comments. Most of these comments refer to the inadequate documentation of the revisions the reviewer suggested in the first round. Questions still persist about what approach to integrative taxonomy was taken, and some issues with the multivariate approach used. I don't believe it is necessary to employ the additional advanced multivariate methods suggested by the reviewer, but these should at least be mentioned as future alternatives. The issue of naming is also important to consider. Please also clarify the sample sizes and undertake the more comprehensive phylogenetic analysis suggested by the reviewer.

Both reviewers recommend discussing the potential for new species in the complex. And both reviewers have a number of minor grammatical errors to fix.

Reviewer 1 ·

Basic reporting

no comment

Experimental design

no comment

Validity of the findings

no comment

Additional comments

Overall, the authors have addressed my concerns regarding the ms. I think some minor changes would improve it further but think it is much improved as is.

I comment about this further in minor comments, but it needs to be explained very clearly what this complex includes for currently named taxa for this complex---make that clear in the intro or methods what the species recognized are!

Also, some discussion as to why the tree has overall such low support in the results/discussion is warranted further.

Around line 76, the methods as to comparing the new species with others start. However, what initial examination happened that was noted to start this “new spp” vs the other Leptophis? It isn’t clear how the new species was noted as new for comparisons (it is alluded to in the previous paragraphs of the intro but not clear whether it was morph or molecular or what that tipped it off).

It would be worth talking about the potential (and why) for additional new species (or not) of Leptophis in the discussion. Eg, nigromarginatus is not monophyletic in the tree---despite avoiding too much talk about the tree due to areas of poor support, it is included as a figure so it’s hard to not want to hear a bit more about it.

Minor:

Line 62 “This comprehensive approach is particularly valuable morphologically similar taxon”
Should read “This comprehensive approach is particularly valuable for morphologically similar taxa”
Line 118---what are the species specific to the ahaetulla complex being tested here?
Line 130---which four individuals/why these four? Are these the new taxon or other species in the complex?
Line 167---I don’t think you are supposed to state the name of the new taxon in the results here prior to the description? I am not sure the way this is done for PeerJ, but it seems odd, especially to not even have the “nov.” part there
Line 171---spelling of ahaetulla
Line 176 and continuing throughout---missing italics for species
Table 1 should state the gene/locus being used

Reviewer 2 ·

Basic reporting

I have reviewed manuscript #98156, a potentially significant contribution to our field. The authors have satisfactorily addressed some of the issues I raised before, demonstrating the importance of their work. Unfortunately, they have not provided a detailed response to my major comments. Therefore, several problems still need to be solved in the new version of the manuscript.

For instance, under “Basic Reporting,” I indicated the need to detail how they have adopted an integrative approach to describing a presumed new species. Even though they added a whole new paragraph to the Introduction and expanded another one on integrative taxonomy, they have not explicitly defined and justified their approach and workflow. For example, Padial et al. (2010), which the authors have cited, illustrated different approaches to integrative taxonomy, e.g., integrating different taxonomic characters by cumulation or congruence. Authors must explain and justify their approach in the Introduction and Material and Methods.

In addition, under “Basic Reporting,” I indicated that the raw data in the manuscript are incomplete and that the authors should provide the complete alignment (not just the newly generated sequences) and the meristic and morphological data of all 1625 examined specimens. Neither the complete alignment nor the meristic and morphological data were provided.

Finally, under “Basic Reporting,” I pointed out that “The figures need many improvements to illustrate differences between sympatric or closely related species and the geographic distribution of species that occur in the region. Authors should refrain from directing readers to other articles to obtain this information.” No change was implemented here. A good description must explain and illustrate how the new species differs from its closely related or sympatric relatives. Regarding the map, authors replied “According to the most recent taxonomic revision of L. ahaetulla (Albuquerque & Fernandes 2022), only L. ahaetulla occurs in the region represented on this map. However, the specimens referred to as L. ahaetulla by these authors (i.e., the CHUNB specimens) were herein reidentified as L. mystacinus. Please check the synonymic list and the referred specimens listed in the present study.” Nevertheless, the map in Nogueira et al. (2019) indicates several other distribution records in the region represented on the map. A good species description should not only represent the geographic distribution of the new taxon but also the distribution of its close relatives. For instance, such geographic distribution information can test or validate species hypotheses drawn from other data types (Puillandre et al., 2012).

Experimental design

Regarding the “Experimental Design,” I stressed the need for multivariate analyses to evaluate the usefulness of the morphological data delimiting the presumed new species. The authors should be commended for adding new analyses to the text, a combination of Principal Components Analysis (PCA) and Multivariate Analysis of Variance (MANOVA). However, it is not clear why they only used “two meristic (SVL, TL) characters and two morphometric (VE, SC) characters” when they had dozens of other characters available (e.g., meristic, morphometric, color pattern, and hemipenial characters of 1625 specimens!). Moreover, the combination of PCA and MANOVA—an indirect approach to discrimination—is inadequate in this context. PCA is an ordination method that summarizes most of the total variation in a multivariate dataset across a reduced number of new dimensions (e.g., linear combinations of the original variables). Yet, the new dimensions do not maximize differences among predefined groups (i.e., species) in the data. A direct approach, such as Discriminant Canonical Analysis (DCA), Support Vector Machines (SVM), Artificial Neural Networks (ANN), and Random Forest (RF), among many others, is preferable and should be used. Considering the different kinds of data (e.g., meristic and morphometric), methods that make fewer assumptions about the data distributions, such as ANN and RF, should be chosen.

Still under “Experimental Design,” I indicated that the phylogenetic analysis should include all available data, including those in Torres-Carvajal, O. & Terán, C. (2021), to produce a robust and up-to-date inference. Nevertheless, authors used only the 16S sequences. Even though they have only obtained 16S sequences of the presumed new species, they should include all available sequences in the analysis to ensure a robust inference. Moreover, I stressed the need for a rigorous, quantitative approach to support the existence of a new species, such as comparing the intra- vs. interspecific genetic distances or the General Mixed Yule Coalescent (GYMC). It is unfortunate that authors over-relied on a single mtDNA marker, as this is not enough to support robust taxonomic decisions. For a recent review on the subject, see Wüster et al. (2024). A quantitative approach, such as GYMC, is essential for the species delimitation workflow (Puillandre et al., 2012).

Regarding the proposition of vernacular names, I thank the authors for their detailed response and clarification regarding their perceived distinction between vernacular and common names. However, the Oxford Dictionary of English indicates:
vernacular
adjective1 (of language) spoken as one's mother tongue; not learned or imposed as a second language.

The Oxford Thesaurus of English indicates:
vernacular
Noun
“he wrote in the vernacular and adopted a non-academic style accessible to the public”
everyday language, spoken language, colloquial speech, native speech, conversational language, common parlance, non-standard language, jargon, -speak, cant, slang, idiom, argot, patois, dialect; regional language, local tongue, regionalism, localism, provincialism;

Further, I would like to emphasize a few essential considerations that should be factored into the naming process, especially in the context of vernacular names.
Vernacular Names as Cultural Artifacts—Vernacular names are not just labels for species but cultural artifacts that emerge organically within communities. These names often carry rich historical, cultural, and linguistic significance, reflecting local knowledge and species relationships. When scientists invent new vernacular names, especially without community input, there is a risk of erasing these cultural connections and imposing an external perspective that may not resonate with the people interacting with these species daily.
Potential for Confusion—While authors argue that inventing vernacular names facilitates scientific communication, it’s essential to recognize that these names might not be adopted or understood by local communities or the broader public. This could lead to confusion rather than clarity, as the invented names may not align with or replace the established common names already in use. This discrepancy could hinder communication rather than enhance it, particularly in conservation efforts where local engagement is crucial.
Importance of Community Involvement—The naming process should ideally involve the communities that are most familiar with the species. Ethnobiological studies can be a valuable approach to documenting the names and meanings that local people already use. These names can then be standardized, if necessary, but they should respect and reflect the knowledge and preferences of those communities. This participatory approach not only honors cultural heritage but also fosters a sense of ownership and responsibility for the conservation of the species.
Precedent in Naming Conventions—The references authors provided demonstrate that there is a precedent for scientists proposing vernacular names. However, it is essential to consider whether those names have been effectively integrated into public use or remain confined to scientific literature. The goal should be to create a naming system that is functional, widely accepted, and respectful of existing cultural practices. This balance can often be better achieved by using or adapting the names that local people already use rather than creating entirely new ones.
Therefore, I recommend revisiting the approach to naming in the manuscript. Instead of inventing new vernacular names, authors should consider conducting a survey or study to identify existing common names for the presumed new species used in regions where it occurs. If no vernacular names exist, creating a new name should involve consultations with local communities to ensure the name is meaningful and likely to be adopted.

Regarding the lack of a consistent sample size in the Results:
ventrals 158–166 in males (x̅ = 160.8 ± 3.6, n = 5), 158–173 in females (x̅ = 167.8 ± 5, n = 8)
Sample size = 5 males + 8 females = 13 individuals

subcaudals 153–164 in males (x̅ = 159.8 ± 4.8, n = 4), 141–158 in females (x̅ = 151.8 ± 8.2, n = 5)
Sample size = 4 males + 5 females = 9 individuals

supralabials 8 (n = 19), 9 (n = 6), or 7 (n = 1), with fourth–fifth (n = 20) or fifth–sixth (n = 6) entering the orbit
Sample size = (19 + 6 + 1) / 2 = 13 individuals

infralabials 10 (n = 17), 9 (n = 7), or 11 (n = 2), with first five (n = 19), four (n = 5), or six (n = 2) contacting first pair of chinshields
Sample size = (17 + 7 + 2) / 2 = 13 individuals

preoculars 1 (n = 26) or 2 (n = 6)
Sample size = (26 + 6) / 2 = 16 individuals

postoculars 2 (n = 25) or 3 (n = 1)
Sample size = (25 + 1) / 2 = 13 individuals

anterior temporal 1 (n = 26)
Sample size = 26 / 2 = 13 individuals

posterior temporal 2 (n = 22) or 1 (n = 4)
Sample size = (22 + 4) / 2 = 13 individuals

Again, please check for consistency! How many individuals were examined?

Regarding the lack of natural history data, authors should at least retrieve whatever info is available from the collections where specimens are deposited or from collectors. I consider the lack of natural history data a significant shortcoming in a species description.

Validity of the findings

Considering all the available data and the scarcity of analyses and results, the evidence for a new species is unconvincing.

Additional comments

Literature Cited
Nogueira, C. C., Argôlo, A. J. S., Arzamendia, V., Azevedo, J. A., Barbo, F. E., Bérnils, R. S., Bolochio, B. E., Borges-Martins, M., Brasil-Godinho, M., Braz, H., Buononato, M. A., Cisneros-Heredia, D. F., Colli, G. R., Costa, H. C., Franco, F. L., Giraudo, A., Gonzalez, R. C., Guedes, T., Hoogmoed, M. S., Marques, O. A. V., Montingelli, G. G., Passos, P., Prudente, A. L. C., Rivas, G. A., Sanchez, P. M., Serrano, F. C., Silva, N. J., Jr., Strüssmann, C., Vieira-Alencar, J. P. S., Zaher, H., Sawaya, R. J. & Martins, M. (2019). Atlas of Brazilian snakes: Verified point-locality maps to mitigate the Wallacean shortfall in a megadiverse snake fauna. South American Journal of Herpetology 14, 1-274.
Padial, J. M., Miralles, A., De la Riva, I. & Vences, M. (2010). The integrative future of taxonomy. Frontiers in Zoology 7,
Puillandre, N., Modica, M., Zhang, Y., Sirovich, L., Boisselier, M., Cruaud, C., Holford, M. & Samadi, S. (2012). Large-scale species delimitation method for hyperdiverse groups. Molecular Ecology 21, 2671-2691.
Wüster, W., Kaiser, H., Hoogmoed, M. S., Ceríaco, L. M. P., Dirksen, L., Dufresnes, C., Glaw, F., Hille, A., Köhler, J., Koppetsch, T., Milto, K. D., Shea, G. M., Tarkhnishvili, D., Thomson, S. A., Vences, M. & Böhme, W. (2024). How not to describe a species: lessons from a tangle of anacondas (Boidae: Eunectes Wagler, 1830). Zoological Journal of the Linnean Society 201, 1-26.

---

## Round 0.3 · Minor Revisions

The reviewer requests additional reporting on aspects of the analysis, including GYMC, random forest models, meristic measurements, and various output statistics of analyses performed. Including the R scripts used is also very important and standard practice these days for any journal.

Reviewer 2 ·

Basic reporting

I still miss figures comparing the diagnostic characters of the presumed new species with closely related or similarly distributed species.

Experimental design

The authors added a GMYC analysis, as suggested, which is most welcomed. However, some details are lacking. First, the “Material and Methods” section should indicate whether they used the single-threshold or the multiple-threshold version of the GMYC and the chosen upper and lower limits of estimation of scaling parameters in SPLITS. Second, they must provide the output statistics, including the log-likelihood ratio test of the fitted model against a null model of no distinct species clusters, and plot the support values (e.g., gmyc.support) of the delineated species on the tree (Supplementary Material 3). These results are essential to evaluate model performance and possible inaccurate delimitations. Authors should also provide the R script and tree file used in these analyses.

The authors have also added a random forest analysis to handle the morphological data. Still, some details are missing. For instance, they must indicate in the “Material and Methods” how many trees were grown. Moreover, in the “Results,” they must indicate the accuracy of each classification in addition to the predictors’ importance. The model accuracy is critical to determine the usefulness of the sampled morphological characters in identifying the presumed new species. Please check the text for instances where “accuracy” is erroneously used instead of predictor importance (e.g., “Although SVL and TL contributed less, the model showed high accuracy for VE and SC, with values of 98.15 and 63.88, respectively.”). The values reported in the text are apparently values of the Gini index and not model accuracy values. Model accuracy should ideally be estimated using k-fold cross-validation instead of the out-of-bag error provided by the package RANGER. Authors should also provide the R script and tree file used in these analyses.

Validity of the findings

The validity of the findings cannot be fully assessed as only some of the underlying data have been provided. Despite my earlier comments, authors must still provide the meristic and morphological data (see above, under "Experimental Design," additional materials that must be provided).

---

## Round 0.4 · accepted · Accept

Thank you for incorporating the last suggestions of the reviewer. I am now pleased to recommend acceptance of your article to PeerJ.